# Pork Meat Composition and Health: A Review of the Evidence

**DOI:** 10.3390/foods13121905

**Published:** 2024-06-17

**Authors:** Filipa Vicente, Paula C. Pereira

**Affiliations:** Applied Nutrition Research Group (GENA), Nutrition Lab, CIIEM—Egas Moniz School of Health & Science, Caparica, 2829-511 Almada, Portugal; pmpereira@egasmoniz.edu.pt

**Keywords:** pork meat, meat, dietary fat

## Abstract

Meat has been part of the human diet for centuries and it is a recognizable source of high-biologic-value protein and several micronutrients; however, its consumption has been associated with an increased risk of non-communicable diseases (e.g., cardiovascular diseases, cancer). These concerns are mostly related to red meat. However, meat composition is quite variable within species and meat cuts. The present study explores the composition of pork meat, and the differences among different pork meat cuts and it reviews the evidence on the influence of its consumption on health outcomes. Pork meat contributes to 30% of all meat consumed worldwide and it offers a distinct nutrient profile; it is rich in high-quality protein, B-complex vitamins, and essential minerals such as zinc and iron, though it contains moderate levels of saturated fat compared to beef. Additionally, research on sustainability points out advantages from pork meat consumption considering that it is a non-ruminant animal and is included in one of the five more sustainable dietary patterns. In what concerns the data on the influence of pork meat consumption on health outcomes, a few clinical studies have shown no harmful effects on cardiovascular risk factors, specifically blood lipids. Several arguments can justify that pork meat can be an option in a healthy and sustainable diet.

## 1. Introduction

Meat has been part of the human diet for centuries and has been an important source of high-quality protein and several other nutrients. Despite this, evidence has shown that meat consumption, specifically red meat, is associated with increased risk of several non-communicable chronic diseases especially cardiovascular diseases and cancer [1,2,3]. This has been used as a reason to advise for a reduced meat consumption in dietary guidelines [4].

However, this recommendation is not free of controversy. Some systematic reviews have shown a small or even null effect of meat consumption in cardiovascular disease or cancer risk and point out several confounder effects that can influence this [5,6,7]. An important variable to consider is the variety of meats included in this classification and their composition.

Pork meat is generally classified as red meat [8] and therefore guidelines suggest its avoidance. However, it is known that meat composition is quite variable among cuts [9], origins and animal nutrition [10,11] as well as preparation techniques [12,13] can have an important role in the final nutritive content and in its impact on several health outcomes. One argument used to reduce meat consumption is its fat content, but it is quite variable among different animal muscle parts and cuts [14].

It is important to also consider factors that justify the preference of consumers for pork meat. Quality, freshness and flavor are relevant arguments for choosing pork meat [15,16]. Cultural aspects are also relevant in meat choice, especially in this case considering that pork meat is not allowed in some beliefs and religions (e.g., Islamism) [17], while in specific regions, pork meat is traditionally the most common option as is the case with countries such as northwestern Spain and northern Portugal. In these geographic regions, dietary practices are closer to the Atlantic diet rather than the Mediterranean dietary pattern [18].

Sustainability is also a relevant subject in what concerns meat consumption. The environmental impact of pork meat has been recently discussed in a review conducted by Drenowski et al. [19], but data on the contribution of different meat options, as well as other protein sources, in terms of nutritive content and also their environmental impact are still scarce. Therefore, it is important to consider the environmental, socioeconomic, nutritional and cultural impact of generalizing meat options that are so different (e.g., lamb, chicken, pork, beef).

The aim of the present study is to analyze and present coherent composition of data on pork meat, establish its possible impact on health outcomes and discuss the presence of this meat option in a sustainable and healthy dietary pattern.

## 2. History Facts and Data on Pork Meat Consumption

Pigs (*Sus scrofa domesticus*) were domesticated from their ancestors, the wild boars (*Sus scrofa scrofa*) in multiple places around the world including Near East, Europe, China and South East Asia 9000 years ago [20]. These animals were easier to hide during enemy attack in ancient times and because of their rapid reproduction and growth, pigs were considered a symbol of prosperity, especially in Europe where an extensive work had developed multiple breeds with a distinguished muscle development for meat production [21].

Since it was domesticated, pigs were genetically improved in order to develop specialized breeds through traditional and assisted selective breeding [22].

About 30% of all meat consumed in the world is pork. The special economic regions of Hong Kong and Macao remained the great pork consumers in 2021, with a consumption of about 56 and 52 kg per capita, slightly higher than Poland, Spain and Germany in Europe (Table 1).

Guenther et al. [24] reported that chicken consumption was associated with a higher income and pork consumption with a lower income. This shows that socioeconomic factors can significantly impact meat consumption behavior [25]. This could also be observed in studies evaluating the effect of the COVID-19 constraints in household income and its impact on animal foods consumption [26], but Milford et al. [27] had previously suggested income as a major driver for meat consumption. Education level can also have an impact on meat intake; individuals with higher levels of education tend to be more aware of the potential health and environmental implications of excessive meat consumption and be more inclined to plant-based or flexitarian diets [27,28].

Some European countries are major pork meat consumers, including Spain, Poland and Lithuania, with pork consumption higher than 40 kg/capita/year [29].

In addition to the lower cost [29,30], cultural aspects can be the reason to justify pork meat consumption, especially in Iberian countries. The preference for pork meat can also be due to cultural aspects in these countries [31] considering the economic importance of the Iberian pork breed [32,33]. Therefore, in these countries, the change from pork meat to poultry or a significant reduction in meat consumption could have deleterious economic effects.

Meat quality is also an important argument for choosing pork and it includes organoleptic properties such as appearance, tenderness, juiciness, aroma and flavor. In addition to these, quality, freshness, price, origin and fat content are important factors for consumers when choosing pork and these are probably the reasons why they prefer fresh and locally produced meat [15,34].

## 3. Nutritional Composition

Meat composition can be affected by several factors including origin, meat cut as well as if it is cooked or raw. Food composition databases show heterogenous data; therefore, it is important to choose a composition table that is continuously updated. In the present paper, the Portuguese food composition table was chosen considering that it is in compliance with the most appropriate European quality management systems and international standards [35].

The energy value of 100 g of pork meat can vary as much as 131 to 355 kcal, and the fat content can be as low as 4.7 g/100 g or as high as 31.8 g/100 g. The protein content is quite similar among cuts but lower in the fattest ones.

### 3.1. Fat Content and Fatty Acid Profile

As presented in Table 2, loin is the leanest meat cut and has a balanced fatty acid profile considering where it supplies similar amounts of monounsaturated fatty acids and saturated fatty acids.

The higher fat content can be found in pork chops which can be cut from different animal parts, and this justifies the fat content, namely if they come from shoulder or leg cuts (Appendix A).

When comparing pork meat cuts with the other most common meat options, pork loin has a lower fat content than beef loin and all pork cuts have higher linoleic acid. If compared with chicken, normally recommended for lower fat content, pork loin and pork leg meat have a lower fat content than chicken meat with skin. Poultry meat skin is indeed a factor to consider when recommending this type of meat for lower fat content [37].

Also, in spite of being higher in fat, chops and ribs also have higher MUFA content (Table 3). In fact, pork loin meat composition is in accordance with the American Heart Association recommendations while beef loin has a higher fat and SFA content [38]. The fattest cut is pork belly, which appears to be one of the favorite cuts in some countries where pork consumption has increased in the last decade in places such as South Korea [39]. Other countries consume this cut mainly in the form of bacon [40].

The fat content and fatty acid profile affects not only the potential effect on health outcomes but also meat sensory characteristics and shelf life. SFAs are crucial for meat texture but have potential hazardous health effects while MUFAs contribute to meat tenderness and flavor [41,42].

Data on meat fatty acid profile are quite variable, while Pleadin et al.’s [43] study revealed that fatty acid profile follows the sequence of MUFAs > SFAs > PUFAs; Covaciu et al. [9] presented a higher average SFA which is also not aligned with data from the present study tables. Nevertheless, the three results confirm that it is interesting to prefer loin cuts considering the lower total fat content and the higher MUFA/SFA ratio compared with other cuts. The second spot is occupied by pork leg, which has one of the lowest fat contents and an interesting MUFA profile, characterized by a significantly higher content of monounsaturated fatty acids (MUFA).

Therefore, the balance between MUFA and SFA in pork leg meat has a major role in striking the consumer preferences for meat flavor and tenderness while ensuring health-focused dietary recommendations.

### 3.2. Micronutrient Composition

In terms of micronutrient content, it is well known that meat is a recognizable source of B complex vitamins, some minerals and trace elements such as iron [10,44]. Pork meat is not an exception; as presented in Table 4 and Table 5, pork meat has a high content of Thiamin, Niacin, Riboflavin and cyanocobalamin as well as phosphorus and it is a source of potassium and zinc.

The composition of pork might explain why its consumption is linked to better nutrient intake and adherence to nutrient recommendations among children (aged 2–18 years) and adults in the United States for several important nutrients. This suggests that pork meat could play an essential role in reducing the prevalence of under-nutrition [45]. A previous study [46] demonstrated that consuming pork, especially lean cuts, provides higher energy-adjusted levels of protein, selenium, thiamin and vitamin B6 compared to diets of adults who do not include fresh pork meat on a given day, while also delivering comparable amounts of total fat and saturated fat.

### 3.3. Special Cuts Composition

In addition to these meat cuts, it is also common in some countries to find specific cuts of pork meat, namely pork belly and liver steaks. The composition of these special pork meat cuts is presented in Table 6.

Liver steaks have a similar fat content to pork loin but are a valuable source of several micronutrients. Liver has a high content of all B complex vitamins, including folates (B9), vitamin A, iron and zinc. The only downside is its cholesterol content (237 mg/100 g) considering that there is still a recommendation for caution on this dietary component, especially in populations at risk. Even so, there is some controversy about the evidence if there are harmful effects from dietary cholesterol in cardiovascular disease risk [47,48,49]. Avoiding liver steaks may result in a reduction in great micronutrient sources such as iron.

Although it is only included in some countries, ears, snout and even the pig’s tail can be included in specialty dishes but there are no full data on their nutritional composition.

Fattier options such as pork belly remain less interesting from the nutritive point of view considering the higher fat and lower protein content as well as a considerably lower micronutrient content.

## 4. Influence of Pork Meat Consumption in Health Outcomes

### 4.1. Cardiometabolic Health

There is a common belief that eating pork meat might increase the risk of cardiovascular disease due to its fat content, particularly its fatty acid composition. However, evidence from randomized clinical trials, as highlighted in Table 7, suggests otherwise. Studies have shown that incorporating lean pork into the regular diet can improve blood lipid profiles, including lowering total cholesterol and LDL cholesterol while raising HDL cholesterol.

For example, Davidson et al. [50] found that a 36-week dietary intervention with lean cuts of red meat, including pork, had a beneficial impact on serum lipid profiles. Similarly, Rubio et al. [51] demonstrated that the consumption of lean pork or veal resulted in similar improvements in lipid profiles among healthy subjects. These findings suggest that both lean pork and veal can be part of the dietary guidelines aimed at controlling saturated fat (SFA) and cholesterol intake.

Pork meat fatty acid profile can possibly justify this, considering that it can be distinguished for its MUFA content and several benefits have been attributed to these fatty acids, specially to oleic acid [52]. Nevertheless, none of these studies include only pork meat in the study group; therefore, we cannot address the benefits of its consumption as it is included as part of a healthy dietary pattern.

**Table 7 foods-13-01905-t007:** A summary of controlled trials published to evaluate the effect of pork meat consumption in cardiovascular disease risk and risk factors.

Reference	Study Features	Result Summary
Davidson et al. [50]	191 men and women LDL 130–190 mg/dL Experimental group was instructed to consume 170 g (6 oz) of meat. 5 to 7 days per week for 36 weeks At least 80% in the form of lean beef. veal. or pork	There were no significant differences in the results produced by the intervention diets in low-density lipoprotein cholesterol and elevations in high-density lipoprotein cholesterol levels
Rubio et al. [51]	44 healthy individuals 6 weeks with 5 weeks for washout Double crossover Veal vs. pork meat (150 g/day)	Lean pork and veal produce similar effects on the lipid profiles of healthy subjects
Hunninghake et al. [53]	N = 145 men and women Hypercholesterolemia 2 × 36 w with 4-week washout phase 170 g red meat/day vs. white	The diet including pork meat was similarly effective for reducing LDL cholesterol and elevating HDL cholesterol concentrations
Stewart et al. [54]	20 adult women Standard pork and lard or the modified pork and lard PUFA enriched pork meat Crossover	The decreases in plasma total cholesterol, LDL cholesterol and SFA contents were most likely a response to the decreased dietary intake of SFAs
O’ Connor et al. [55]	41 subjects 2 × 5 weeks MedDiet. one of 2 versions: MedRed vs. MedControl 500 g vs. 200 g red meat/week 4 weeks washout between	Total cholesterol decreased, greater reductions occurred with MedRed than with MedControl
Wade et al. [56]	31 Adults 45–80 years old A 24-week parallel crossover design trial MD intervention with 2–3 weekly servings of pork (MedPork) with an LF control intervention	No significant differences were observed
Montoro-Garcia et al. [57]	54 volunteers with stage 1 prehypertension and/or hypercholesterolemia and/or basal glucose >100 mg/dL 80 g cured ham with added bioactive compounds 2 × 4 weeks with a 2 week washout	Total cholesterol levels also decreased significantly after dry-cured ham intake

LDL: low-density lipoprotein; HDL: high-density lipoprotein; SFA: saturated fatty acids.

As shown, few studies had evaluated the influence of pork meat in blood lipids and/or cardiovascular risk factors, and none had included only pork meat. Despite clinical studies and especially randomized clinical trials being essential to build up robust evidence on the effect of a compound, a food group or several foods on disease risk factors or health outcomes, this remains challenging in nutrition sciences considering that the human diet is complex [58], there are multiple food-to-food and nutrient-to-nutrient interactions. People do not eat nutrients or isolated foods and dietary patterns where the foods of interest are included/excluded can have a higher similarity to the complex reality of human food habits in these studies [59].

### 4.2. Body Weight, Obesity and/or Adiposity

The systematic review and meta-analysis conducted by An and colleagues [60] found null or inverse associations between pork meat consumption and body weight and/or body fat. The authors reported that in spite of red meat consumption being previously associated with weight gain and abdominal adiposity [61], most studies did not consider bias such as other unhealthy food and lifestyle habits of regular red meat eaters.

Nevertheless, as presented in Table 8, few randomized clinical trials have specifically addressed pork meat and their results have shown that lean pork meat consumption is a viable protein source in an energy restricted meal plan for weight and body fat loss. In fact, the study conducted by Murphy et al. [62] has shown no significant differences in body mass index and fat mass between the three dietary interventions with chicken, pork or beef meat. In Campbell and Tang’s study [63], the authors report that no differences were found in BMI, weight and fat mass loss within diet protocols, despite there being significant differences between the before and after study values which was also shown by Murphy et al. [62].

Protein has a crucial role in weight control considering that it acts in satiety, energy expenditure as well as preserving fat-free muscle, and thus its benefits are also on body composition [64,65]. Considering this, pork meat can be an affordable protein source in a weight loss diet, especially lean meat cuts. Nevertheless, energy restriction is the main factor for a weight loss diet therapy protocol and despite this being a unique strategy [66], reducing dietary fat has shown interesting results; therefore, lean pork meat cuts can be an option.

**Table 8 foods-13-01905-t008:** Randomized clinical trials on pork meat consumption and its effect on body weight and/or body fat or BMI.

Reference	Study Features	Result Summary
Mikkelsen et al. [67]	N = 12, only men RCT BMI 26–32 4-day isoenergetic intervention 3-way crossover (1) Low-fat, high pork-meat protein diet (pork diet); (2) Low-fat, high-soy-protein diet (soy diet); (3) Low-fat, high-carbohydrate diet (carbohydrate diet)	There were no differences in body weight between the three protocols
Campbell W. & Tang, M. [63]	N = 28, only women 12-week 750 kcal/d energy-deficit diet containing higher or normal protein In the high protein group, 40% was from pork meat Normal diet was egg-lacto-vegetarian	Postmenopausal women in both NP and HP (40% pork) energy restriction diet groups showed decreases in BMI, fat mass and lean mass (*p* < 0.001); however, no difference was found between normal protein and higher protein diet on BMI, fat mass and lean mass
Murphy et al. [62]	N = 49, adults 140 g/day chicken, 150 g pork or beef Crossover design: 3 months, 1-week washout	There was no difference in BMI, body fat percentage, fat mass, abdominal fat, lean mass, WC and HC when comparing pork group with beef or chicken diet group (*p* > 0.05); WHR was lower in pork group than beef and chicken group (*p* = 0.046)

BMI: body mass index; WC: waist circumference; HC: hip circumference; WHR: waist-to-hip ratio.

### 4.3. Pork Meat Consumption and Cancer Risk

Although there is some heterogeneity, no associations have been found between pork consumption and colorectal cancer risk in opposition to the possible effect of beef or lamb composition [68]. One possible explanation for this would be the lower content of iron in pork meat. Gamage et al. [69] suggested that heme iron, present in meat, can have a dual effect in colon carcinogenesis. Despite this, it could suppress tumors by ferroptosis, and data showed it can modify immune cell function, promoting inflammation and gut dysbiosis, inhibiting the tumor suppressive potential of the P53 gene, enhancing cellular cytotoxicity and reactive oxygen species formation.

For other cancer sites, Zhu et al. [70] found no significant association between pork consumption and gastric cancer risk. Nevertheless, for breast cancer, pork meat consumption is not clearly identified as a risk factor in most studies; normally it is included with red meat, and some studies refer to the deleterious effects of processed meats [1,70].

It is important to highlight the vast complexity of cancer etiology; despite this, diet has been associated with 30% of cancer cases, but there are several bias factors to consider when studying the effect of diet patterns, specific foods and food components in cancer risk. In what is concerned with meat consumption, there are scarce data about different meat cuts and the meat composition being influenced by its origin and production methods including animal diet manipulations [71].

Dietary fat is a possible risk factor for cancer [72,73,74]; however, it is not free of controversies [75]. Pork meat is commonly avoided because of fat content, but as previously presented, there are leaner options.

### 4.4. Other Health Outcomes

There is no evidence on the detrimental effects of adequate/moderate pork meat consumption in any other health outcomes. For instance, Datlow et al. [76] have shown that pork does not have any benefit, nor does it have a harmful influence on cognitive performance.

In what is concerned with diabetes risk, evidence does not even support any effect from red meat itself [77].

There are major challenges when considering the effect of one single food in disease risk or any health outcome. It is not possible to ensure that a clinical trial is blind in what comes to food intake considering that it would be necessary to restrict a specific food in the placebo group. Therefore, most studies on pork meat consumption include it in a specially designed nutritional intervention for the study. One possible suggestion is to include the studied food in a defined dietary pattern, such as defining specific weekly frequencies of pork meat consumption, and study the adherence to this pattern and the risk of diseases or other health outcomes, which has been the case with the Mediterranean diet but not with the referenced Atlantic diet where pork meat would be included [78].

## 5. Pork Meat in a Sustainable and Healthy Dietary Pattern

The Mediterranean diet is the most frequently recommended dietary pattern considering the multiple benefits in promoting health and chronic disease prevention. It advises a low intake of red meat; therefore, pork would not be included. However, another dietary pattern has emerged in the Iberian countries—the Atlantic diet or the South European Atlantic diet that is traditional from northwestern Spain and northern Portugal. In these regions, pork is a stable food together with dairy, legumes and vegetables, as well as specific seafood species like cod and octopus [18,79]. Recently, a high adherence to the Atlantic diet has been associated with lower all-cause, cardiovascular and cancer mortality as well as with a low depression risk [80,81,82]. Despite this, authors did not ignore that this raises some questions considering that the advice is inconsistent with health recommendations and in one of their studies, the consumption of meat was reversely scored [80]. Nevertheless, the moderate consumption of meat in this dietary pattern comes from autochthonous bovine and porcine breeds from extensive livestock farms, where the animals are fed based on grass and milk, in the case of veal (Galician blonde calves and Cachena breed calves), and with chestnuts in the case of pork (Galician Celtic pig) which may lead to different effects in their composition and nutritive value [83]. As presented by Lebret and Čandek-Potokar [84], production factors have a major impact on pork meat quality.

In these regions, the common recommendation to avoid meat consumption would have detrimental socioeconomic consequences. In the Mediterranean region, pork meat consumption tended to be lower than other European countries [85] which could be explained considering the Muslim influence in this area. Also, in the Mediterranean Sea-bathed areas, there is an easier access to fresh fish but this would not be the case in interior areas. For this reason, some southern regions of Portugal like Alentejo also have an important history on pork meat production as well as in specific culinary recipes for it preparation [86].

The meat consumption environmental impact can also be questioned in this case. Considering that the carbon footprint is one of the most relevant variables in sustainability assessment, pork meat production systems have lower carbon footprints than beef [87,88].

Nevertheless, most recommendations suggest a decrease in meat consumption for environmental reasons, and that a sustainable and healthy diet should be adapted to local conditions. In these regions, replacing pork meat by other protein sources (e.g., fish) would not take into consideration traditionally and culturally established products and would imply the need to import products from other regions which also has an impact on the carbon footprint [89].

Although most data on sustainability suggest that there are clear benefits on removing meat from diet for environmental reasons, a systematic review conducted by Aleksandrowicz et al. [90] showed that meat is not all equal. In this review, a dietary pattern preferring meat from non-ruminant animals, where pork is included, can be considered sustainable together with a vegan or a vegetarian dietary pattern.

## 6. Discussion

The present study aimed to review pork meat composition, establish possible associations with health outcomes and propose a place for pork meat in a healthy and sustainable dietary pattern.

Based on the food composition, there is some controversy when recommending pork meat avoidance based on fat content considering that it is very heterogenous according to the chosen meat cut and this has been reported in meat varieties [91,92]. Also, in the pork meat fatty acid profile, it is possible to highlight the monounsaturated fatty acid content. These fatty acids have been associated with multiple health outcomes and reduced overall mortality risk [93,94]. This can justify why the few clinical studies evaluating the effect of pork meat consumption in health outcomes have shown that it has no deleterious effects on blood lipids, cardiovascular health or even body weight and fat mass [51,54,95,96].

Several factors can also be confounders in the possible influence of meat consumption on human health or disease risk. One of these factors is the processing degree considering that the most robust associations for cancer risk are related to processed meat and not meat in nature [1,5]. This is especially important when referring to the risk of specific cancer forms associated with dietary factors such as colorectal [97] and breast [98] cancer. There is a clear lack of robust evidence comparing the effect of different red meat varieties (pork, beef, poultry) and different degrees of processing in cancer risk [7,99]. Also, as previously discussed in Section 4.1, the complexity of human dietary habits generates multiple biases. Additionally, most data on the effect of foods and food groups in disease risk come from observational studies, which, although being well designed, raise questions when considering the causality and have multiple biases that are not controllable (e.g., lifestyle) [100].

These considerations do not support an insufficient consumption of swine meat; rather, they confirm that, as demonstrated in the few clinical studies in which pork meat was included and showed advantages, it can be incorporated in a healthy diet.

Additionally, it is important to consider that this recommendation to reduce or even remove red meat, and in this case, pork meat from human diet, it is not as sustainable as it is promising. As reported, there is another dietary pattern with benefits in human health characteristic from Iberian countries—the Atlantic diet and our South European diet. This pattern has also been associated with multiple health outcomes [80,81,82] and includes pork meat, because it is common in regions where this livestock is frequent [101].

In addition to the local criteria, pork meat production has been suggested to have a lower environmental impact than beef or even lamb [19,102] despite the possible consequences from the growing market of pork meat worldwide [103].

It is important to consider some limitations in the present review. Data on the pork meat composition considered a specific food composition database. To the best of our knowledge, it was the only database with the complete information on all the considered nutrients considered for analysis: fatty acid composition (MUFA, PUFA and SFA), linoleic acid, vitamins and minerals. Additionally, it was also the only viable source of information for all the available meat cuts. Meat composition, especially in what refers to fatty acids, can vary according to origin and animal nutrition but there is a lack of robust studies presenting these differences with a comprehensive nutritive analysis.

## 7. Concluding Remarks

The presented study reviewed the differences in pork meat cuts’ nutritive composition and the data have shown that pork meat is an affordable source of protein and several micronutrients. Fat content is quite variable among different meat cuts, loin is the leanest option and is in accordance with the dietary guidelines. These data probably justify why the few clinical trials conducted with pork meat have shown null or even inverse associations between its consumption and different risk factors, and specifically when included in a healthy dietary pattern. Additionally, considering the relevance of the sustainability subject, it is crucial to consider that including dietary patterns with pork meat can be considered sustainable. Therefore, as expected, the Atlantic diet that has been previously associated with multiple health outcomes, is also relevant as part of a healthy dietary pattern with attributed benefits to health.

## Figures and Tables

**Table 1 foods-13-01905-t001:** Pork meat consumption per capita in several countries and economic regions [23].

Country	Kg/Capita/Year
China–Hong Kong	55.9
Poland	54.9
Spain	52.6
China–Macao	52
Lithuania	50.7
Germany	44.0
Belarus	39.2
Portugal	38.0
China–Mainland	35.3
France	31.0
United States	30.6
United Kingdom	24.0
Brazil	17.7

**Table 2 foods-13-01905-t002:** Energy, protein, fat and fatty acid content of main pork meat cuts. Data are presented for 100 g of meat [36].

Pork Meat Cut	Energy Value (kcal)	Protein (g)	Fat (g)	SFA (g)	MUFA (g)	PUFA (g)	LA (g)
Pork loin	131	22.2	4.7	1.6	1.6	0.8	0.7
Pork chops	288	18.6	23.8	8.2	7.9	3.4	0.1
Pork ribs	190	19.6	12.4	4.2	4.1	1.7	0.1
Pork leg	190	12.1	12.3	6.3	6.2	2.6	0.1
Pork belly	518	9.3	53	19.3	24.7	5.6	0.0

**Table 3 foods-13-01905-t003:** Comparison between pork meat composition and other meats. Data are presented for 100 g of meat [36].

Meat Cut	Energy (kcal)	Fat (g)	SFA (g)	MUFA (g)	PUFA (g)	LA (g)
Beef, loin	174.5	10.3	3.9	4.6	0.4	0.3
Chicken, no skin	110	2	0.5	0.7	0.4	0.4
Chicken, skin	201	13.6	13.6	3.2	4.5	2.8
Pork loin	131	4.7	1.6	1.6	0.8	0.7
Pork ribs	190	12.4	4.2	4.1	2.1	1.7
Pork chops	288	23.8	8.2	7.9	3.9	3.4
Pork leg	189	12.3	6.3	6.2	2.6	0.1

**Table 4 foods-13-01905-t004:** Vitamin B complex content in pork meat. Data are presented for 100 g of meat [36].

	B1 (mg)	B2 (mg)	B3 (mg)	B6 (mg)	B9 (mg)	B12 (mg)
DRV ^1^	1.1	1.4	16	1.4	200	2.4
Pork loin	0.7	0.2	5.3	0.4	5.0	1.0
Pork chops	0.7	0.2	6	0.4	4.5	1.0
Pork ribs	0.7	0.3	7.2	0.4	1.0	1.0

^1^ DRV = dietary reference value.

**Table 5 foods-13-01905-t005:** Specific mineral and trace element composition of pork meat cuts [36].

	Potassium [mg]	Calcium [mg]	Phosphorus [mg]	Magnesium [mg]	Iron [mg]	Zinc [mg]
DRV ^1^	2000	800	700	375	14	10
Pork loin	400	7	220	23	0.6	1.6
Pork chops	330	15	190	19.5	1.05	1.9
Pork ribs	350	11	190	21	0.8	2.2
Pork leg	395	16	245	18	1.4	1.9

^1^ DRV = dietary reference value.

**Table 6 foods-13-01905-t006:** Energy, fat and protein content of liver and special cuts in pork meat [36].

	Energy [kcal]	Fat [g]	SFA [g]	MUFA [g]	PUFA [g]	LA [g]	Protein [g]
Ear	128	2.5	0.8	0.9	0.8	0.8	26
Liver steak	129	5	1.7	1.7	0.8	0.7	20.9
Pork belly	682	72	24.1	27.9	11	9.4	8.4

## Data Availability

No new data were created or analyzed in this study. Data sharing is not applicable to this article.

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
