# Peer review of "Pork Meat Composition and Health: A Review of the Evidence"

_foods, 2024, doi:10.3390/foods13121905_

Round 1

Reviewer 1 Report

Comments and Suggestions for Authors

Hi dear Editorial board and the respected authors

This article "Pork meat composition and health: a review of the evidence” was revised and has a novelty and I recommend it has not choice for publication even after consideration of the following comments.

Abstract:

·         The type of statistical design used in this research should be mentioned.

·         What are the main distinctions in the composition of red meat, pig, and other meats, and how do they affect human health?

·         How does pork's nutritional profile, including B-complex vitamins and other micronutrients, compare to other protein sources, and what are the consequences for different demographic groups?

·         What are the sustainability benefits of pig consumption over ruminant meats, and how does this relate to suggested dietary patterns?

·         What are the most current results on the particular health implications, both good and bad, of consuming hog meat in human populations?

·         How do dietary recommendations for red meat and pig intake compare or differ, and what is the reasoning behind these guidelines? What are the potential mechanisms?

Introduction:

·         What are the main nutritional and compositional differences between pig and other varieties of red meat, and how do they affect the possible health consequences?

·         How do processing procedures, such as preparation methods and fat content, affect the nutritional profile and health consequences of hog consumption?

·         Given the "conflicting evidence" on the health effects of meat eating, what are the hypothesized processes by which hog meat may increase or decrease risk of chronic illnesses such as cardiovascular disease, diabetes, and cancer?

·         How do consumer views and attitudes regarding hog meat, influenced by information and messaging, compare to scientific knowledge about its health effects? What variables contribute to the possible disconnect? What are the limits of the present study on hog meat and health?

Results:

·         What is the difference between the tables 3 and 4?

·         Table 3. Comparison between pork meat composition and other common meat cuts. Is it right?

·         Table 5. – Specific Mineral and trace elements composition of pork meat cuts. For how many meat?

·         Tables 7 and 8 should be given as text or at least modified to a more correct format

·         Line 128 etc.: please correct the reference citation accordingly.

·         I think combine tables 2, 3, and 6. Of course it is my suggestion and the authors decide which approach is better.

·         It should be noted in the tables how much pork these amounts are

Discussion:

Discussion text must grammar improve and in some cases it is very weak and maybe there is no discussion at all.

Conclusions:

Conclusion is very general, try to make it more scientific, comprehensive and concise in detail, especially.

References: It is OK.

The article has many flaws in express and concept of English, it is suggested to be revised in a scientific and native way.

Comments on the Quality of English Language

The article has many flaws in express and concept of English, it is suggested to be revised in a scientific and native way.

Author Response

This article "Pork meat composition and health: a review of the evidence” was revised and has a novelty and I recommend it has not choice for publication even after consideration of the following comments.

The authors thank the reviewer for the attention given to the paper and the opportunity to improve its contents.

Abstract:

  • The type of statistical design used in this research should be mentioned.
  • What are the main distinctions in the composition of red meat, pig, and other meats, and how do they affect human health?
  • How does pork's nutritional profile, including B-complex vitamins and other micronutrients, compare to other protein sources, and what are the consequences for different demographic groups?
  • What are the sustainability benefits of pig consumption over ruminant meats, and how does this relate to suggested dietary patterns?
  • What are the most current results on the particular health implications, both good and bad, of consuming hog meat in human populations?
  • How do dietary recommendations for red meat and pig intake compare or differ, and what is the reasoning behind these guidelines? What are the potential mechanisms?

The abstract had been rewritten, we appreciated and considered these comments.

Introduction:

  • What are the main nutritional and compositional differences between pig and other varieties of red meat, and how do they affect the possible health consequences?
  • How do processing procedures, such as preparation methods and fat content, affect the nutritional profile and health consequences of hog consumption?
  • Given the "conflicting evidence" on the health effects of meat eating, what are the hypothesized processes by which hog meat may increase or decrease risk of chronic illnesses such as cardiovascular disease, diabetes, and cancer?
  • How do consumer views and attitudes regarding hog meat, influenced by information and messaging, compare to scientific knowledge about its health effects? What variables contribute to the possible disconnect? What are the limits of the present study on hog meat and health?

Thank you very much for the highlights given. The introduction had been rewritten incorporating some of these suggestions and other reviewers.

Results:

  • What is the difference between the tables 3 and 4?

Table 3 presents the vitamin, specifically B complex vitamins, content while table 4 presents the pork meat content for several minerals & trace elements.

  • Table 3. Comparison between pork meat composition and other common meat cuts. Is it right?

We appreciate your comment and changed the table caption.

  • Table 5. – Specific Mineral and trace elements composition of pork meat cuts. For how many meat?

And  It should be noted in the tables how much pork these amounts are:

Thank you for your comment, we added the information in all the tables, data is presented for 100g of meat (3.5oz).

  • Tables 7 and 8 should be given as text or at least modified to a more correct format

We appreciated your comment and suggestion. Additional comments on the referred studies were added in liens 219-222.  

Also, a comment on data presented in these tables had been added through lines 200-208.

  • Line 128 etc.: please correct the reference citation accordingly.

Thank you very much for the attention given to these details. It was corrected.

  • I think combine tables 2, 3, and 6. Of course it is my suggestion, and the authors decide which approach is better.

We understand your suggestion, we kept it divided considering that energy, fat and protein and micronutrient content are separately discussed in each subsection.

Discussion:

Discussion text must grammar improve and in some cases it is very weak and maybe there is no discussion at all.

We agree with the reviewer. The entire manuscript has been revised for linguistic corrections.

Conclusions:

Conclusion is very general, try to make it more scientific, comprehensive and concise in detail, especially.

We agree with the reviewer, this section was rewritten (lines 301-310).

Reviewer 2 Report

Comments and Suggestions for Authors

You can see some remarks in the attached files

Most important is the analysis you present which does not include variations.

Comments on the Quality of English Language

You can see some remarks in the attached files

Author Response

Thank you very much for the attention given to all the manuscript. Your considerations have been very useful to improve our manuscript grammar and vocabulary.

Line 7: change and include to including

The abstract had been rewritten according to other reviewers’ considerations.

Line 9: change Pork to Pork meat

Line 10: contributes to it intake in special populations. Improve language

Line 17: Look at this writing suggestion: Although meat has been part of the human diet for centuries

Line 34: scientif names should be written in italics: Sus scrofa domesticus

The referred corrections had been done, thank you very much for your careful reading.

Line 47: Table 1; I cannot see in the reference the specific table, I believe it will be better to use an alphabetical order or by decreasing amount

We reconsidered and reviewed the references; data had been extracted from FAO Balance sheets and not OCDE information therefore we added another reference. 

Line 64: what is that C after [15,16]?

It was a typing error, thank you very much.

Line 76: Reference is about Portuguese Food Composition Database Quality

Management System and not about the analysis of pork meat.

In this comment we justified the option for the Portuguese Food Composition Database Quality Management System as the source of food composition.

Line 87: where this data is coming from? Are they valid? What is the standard deviation for each parameter? For example: in lines 83-85, using data from table 2 you are saying that loin supplies similar amounts of monounsaturated fatty acids and

saturated fatty acids although it is well known that nutrition plays a significant role in that. The same questions for all the tables presented.

We acknowledge the reviewer’s comment. The presented data was obtained from the PORTFIR food composition table. To our best knowledge is a complete, comprehensive and coherent food composition database and when comparing data from several other, it was the only with data for all the considered nutritive components considered in this review.

We added a comment on this as a subsection about limitations about food composition analysis (Lines 307-315).

Line 113: need to improve language

Line 119: In what regards to micronutrient content suggest to change to: In terms of micronutrient content

Line 121: change by to as

Thank you very much, it was corrected.

Line 122: As you use the full name of each vitamin you should do the same with vitamin B12

Line 122: Tiamin; Line 129: thiamin; Check spelling and choose one.

Based on your comment we decided to use the vitamin names.

Line 130-131: probably wrong cut of the sentence.

Thank you for the comment, it was a wrong typing error.

Line 160: interesting in; change to interesting from

Line 170: Table 6; change to Table 7

Line 195: Table 7; change to Table 8 

Thank you, these lines were all corrected.

Reviewer 3 Report

Comments and Suggestions for Authors

Detail comments:

Abstract:

1. This part is less than a hundred words, and should be supplemented to fully describe the purpose and significance of this review and properly reflect the content of the review.

Introduction:

2. This section is short, please develop some sentences with proper and recent references. For example, in the first paragraph, the research background and current situation of the harm of red meat should be added; in the second paragraph, the policies of many parts of the world (such as Europe, North America, Asia and other countries and regions) that restrict red meat should be added; and in the third paragraph, a summary paragraph of the specific content of this paper should be added.

3. L23, what is the meat composition is quite variable among cuts and preparation techniques can have an important role in the final effect. It is suggested to expand the discussion appropriately and cite relevant references to show its importance.

4. L27-29, please give some recent examples describing the conflicting evidence of meats in health and disease risk.

5. The aim was not clear, please rewrite it.

History facts and data on pork meat consumption:

6. L34, italic form of Latin name.

7. L34-L42, it is suggested to add descriptions of pig origins, domestication, hybridization, morphology, regional distribution, and major breeds.

8. In the table 1, if authors can sort the data by consumption.

Nutritional composition:

9. Sub section 3.1, necessary information should be indicated in the form, such as data source, data year, pork varieties, etc.

10. Please correct the form of the table.

11.In the table 3, Please explain the choice of chicken(no skin) for comparison.

Influence of pork meat consumption in health outcomes:

12. L195-197, missed discussion in this part, please add necessary analysis.

13. Sub section 4.2 and 4.3, this part was not strong enough, please rewrite it.

Pork meat in a sustainable and healthy dietary pattern:

14. It is recommended to discuss the world pork dietary patterns to highlight the role of pork consumption in healthy dietary patterns.

Concluding remarks:

15. This section is weak and should be rewritten carefully.

References:

16. More references should be cited to enhance the in-depth discussion.

Comments on the Quality of English Language

1. The English of this paper needs extensive revision.

2. Please carefully check the full text, spelling and use of professional names. For example, lines 27 -29, line 64.

Author Response

 The authors wish to thank the reviewer for the comments which have unequivocally helped to improve the manuscript.

Detail comments:

Abstract:

  1. This part is less than a hundred words, and should be supplemented to fully describe the purpose and significance of this review and properly reflect the content of the review.

The abstract had been rewritten. 

Introduction:

  1. This section is short, please develop some sentences with proper and recent references. For example, in the first paragraph, the research background and current situation of the harm of red meat should be added; in the second paragraph, the policies of many parts of the world (such as Europe, North America, Asia and other countries and regions) that restrict red meat should be added; and in the third paragraph, a summary paragraph of the specific content of this paper should be added.
  2. 3L23, what is the meat composition is quite variable among cuts and preparation techniques can have an important role in the final effect. It is suggested to expand the discussion appropriately and cite relevant references to show its importance.
  3. 4L27-29, please give some recent examples describing the conflicting evidence of meats in health and disease risk.
  4. The aim was not clear, please rewrite it.

We acknowledge the reviewers suggestions and the introduction was rewritten and the aim was better clarified.

History facts and data on pork meat consumption:

  1. 6. L34, italic form of Latin name.

It was corrected.

  1. L34-L42, it is suggested to add descriptions of pig origins, domestication, hybridization, morphology, regional distribution, and major breeds.

We understand the comment but this would be out of our subject in nutritional point of view considering that there is a lack of references in terms of significant differences in composition, to our best knowledge. 

  1. In the table 1, if authors can sort the data by consumption.

We agree with the reviewer and corrected the table.

Nutritional composition:

  1. Sub section 3.1, necessary information should be indicated in the form, such as data source, data year, pork varieties, etc.

We agree with the reviewer and added a reference for each composition table (Tables 3,4, 5 and 6). Additionally the authors added a paragraph about the limitations considering the source of this information (L308-315).

  1. Please correct the form of the table.

Thank you very much. We revised the template in order to ensure that the table formats were according to the template, we suppose that could be improved if needed.

11.In the table 3, Please explain the choice of chicken(no skin) for comparison.

We understand and appreciate the reviewers comment. We added chicken with skin to the table 3 and a comment (lines 103-106) comparing both. Normally poultry meat is considered a good option for lower fat content reasons but as comparison shows, depends on the detail if it is skinless or not. Because our focus is pork meat, we did not compared other meat cuts rather the average composition of whole chicken but it is known that breast has even a lower fat content which affects organoleptic factors.

Influence of pork meat consumption in health outcomes:

  1. L195-197, missed discussion in this part, please add necessary analysis.
  2. Sub section 4.2 and 4.3, this part was not strong enough, please rewrite it.
  3. It is recommended to discuss the world pork dietary patterns to highlight the role of pork consumption in healthy dietary patterns.

Thank you for these comments, we decided to add a discussion subsection to manuscript.

Concluding remarks:

  1. This section is weak and should be rewritten carefully.

The section was rewritten.

References:

  1. More references should be cited to enhance the in-depth discussion.

A discussion subsection had been added to reinforce the questions raised in some topics.

Reviewer 4 Report

Comments and Suggestions for Authors

Manuscript foods-3034665, entitled “Pork meat composition and health: a review of the evidence

Recommendation:       The above paper is not suitable for publication in its present form.

The article provides useful information about the relationship between pork meat composition and health status of the consumers.

In general, the language used is poor. Several parts should be rephrased. For example, please check L9-10, 18-19, 24, 41, 56-57, 95, 113-114, 119,148-149, 156-158, 166-167, 170, 192-193, 208, 227-228, 271, 277-278 etc

In Tables 2-6, please add references. In Tables 7-8, please remove the references from the first to the last column. In general, please check author guidelines regarding Tables format and explain abbreviations used as a footnote under the Table.

Also, the part of discussion is inadequate. Please enrich your article with up-to-date literature

L62-63: Repetition. Please delete

L97-98: “The fattest cut is pork belly”. Where is this shown?

L121: “…in Tables 4 and 5, average…”

L122: “soure”

L130: “who do not consume”?

L170: Table 6 or 7?

L195: Table 7 or 8?

Author contributions, Data availability Statement, Acknowledgements are not provided

Comments on the Quality of English Language

Extensive editing of English language required

Author Response

The authors thank the reviewer for the careful attention given to the manuscript, the comments and suggestions had improved the quality of our work.

In general, the language used is poor. Several parts should be rephrased. For example, please check L9-10, 18-19, 24, 41, 56-57, 95, 113-114, 119,148-149, 156-158, 166-167, 170, 192-193, 208, 227-228, 271, 277-278 etc

We agree with the reviewer, there were several typing errors. The authors proceeded with a linguistic revision.

In Tables 2-6, please add references.

The reference to the food composition table was added and the justification for this option had been previously included in lines 80-86.

In Tables 7-8, please remove the references from the first to the last column. In general, please check author guidelines regarding Tables format and explain abbreviations used as a footnote under the Table.

Thank you for your recommendation, the abbreviations were included in footnotes under the table.

Also, the part of discussion is inadequate. Please enrich your article with up-to-date literature

A discussion section had been added to highlight some topics.

L97-98: “The fattest cut is pork belly”. Where is this shown?

Thank you for your comment. Data on pork belly composition fat content is presented in Table 2.

L121: “…in Tables 4 and 5, average…”

L122: “soure”

L130: “who do not consume”?

L170: Table 6 or 7?

L195: Table 7 or 8?

Thank you for the attention given to these details, the authors revised all these lines.

Author contributions, Data availability Statement, Acknowledgements are not provided

This does not apply to our manuscript in our point of view. Both authors contributed equally and independently to the manuscript. We added the reference for data availability statement.

Round 2

Reviewer 2 Report

Comments and Suggestions for Authors

See some notes in the attached file

Comments on the Quality of English Language

Author Response

Thank you very much once again for the careful attention given to our work. We corrected according to each indication in this revised version.

About the text in line 120, pork steaks cut from leg and with bone are commonly mentioned as leg pork chops.

Reviewer 3 Report

Comments and Suggestions for Authors

1. Please reduce the frequency of a single sentence as a paragraph if it is not necessary.

2. L84-86 give some recent examples describing the relationship of socioeconomic factors and meat consumption behaviour.

3. In table 8,Kindly provide more examples and discussion.

4. L256-258,Please discuss why the results for pork are opposite to those for beef and mutton.

5.L283-284, what is the one possible suggestion is to include the studied food in a defined dietary pattern.

Comments on the Quality of English Language

Please carefully check the grammar of full text, it is recommended to ask native speakers to modify the full grammar.

Author Response

The authors appreciate the attention given to our revised version and addressed answers to the additional comments which enriched our text and discussion as well as further research topics. 

  1. Please reduce the frequency of a single sentence as a paragraph if it is not necessary.

Text was revised and this was reduced.

  1. 2L84-86 give some recent examples describing the relationship of socioeconomic factors and meat consumption behaviour.

Some more examples were added

  1. 3. In table 8,Kindly provide more examples and discussion.

It was added , thank you for the constructive point of view

  1. L256-258,Please discuss why the results for pork are opposite to those for beef and mutton.

The authors acknowledge the comment and added the most plausible explanation (L273-L278).

5.L283-284, what is the one possible suggestion is to include the studied food in a defined dietary pattern.

We added a further explanation (L305-307)

Reviewer 4 Report

Comments and Suggestions for Authors

Authors made the majority of the suggested corrections.

Please remove the column of "References" at the end in Tables 7 and 8. Some moderate corrections regarding the language of the article should be made 

Comments on the Quality of English Language

Some moderate corrections regarding the language of the article should be made 

Author Response

Thank you very much for the careful attention given to the revised version. 

We acknowledge the comment from the reviewer and added the authors name to be possible to identify the study in each line.